# Risk of Nephrolithiasis in Patients with Inflammatory Bowel Disease Receiving Biologic Treatment

**DOI:** 10.3390/jcm12196114

**Published:** 2023-09-22

**Authors:** Zakaria Alameddine, Racha Abi Melhem, Reem Dimachkie, Hussein Rabah, Hamed Chehab, Michel El Khoury, Faris Qaqish, Dimitre Stefanov, Suzanne El-Sayegh

**Affiliations:** 1Department of Internal Medicine, Staten Island University Hospital, 475 Seaview Avenue, Staten Island, NY 10305, USA; rabimelhem@northwell.edu (R.A.M.); rdimachkie@northwell.edu (R.D.); hussein-rabah@hotmail.com (H.R.); hchehab@northwell.edu (H.C.); melkhoury@northwell.edu (M.E.K.); fqaqish@northwell.edu (F.Q.); selsayegh@northwell.edu (S.E.-S.); 2Biostatistics Unit, Feinstein Institutes for Medical Research, 350 Community Dr, Manhasset, NY 11030, USA; dstefanov@northwell.edu

**Keywords:** nephrolithiasis, inflammatory bowel disease, biologics, vedolizumab, urolithiasis, stone

## Abstract

Introduction: Inflammatory bowel disease is a chronic inflammatory disorder of the gastrointestinal tract. Biologic drugs target specific molecules in the body’s immune system to control inflammation. Recent studies have suggested a potential link between their use and an increased risk of nephrolithiasis. We conducted a study to further investigate this association. Methods: The study used multiple logistic regression analysis to assess the association between the use of biologic drugs and nephrolithiasis. A *p*-value of <0.05 was considered statistically significant. SAS 9.4 was used for statistical analysis. Results: The final sample consisted of 22,895 cases, among which 5603 (24.51%) were receiving at least one biologic drug. The biologic drugs received were as follows: Adalimumab 2437 (10.66%), Infliximab 1996 (8.73%), Vedolizumab 1397 (6.11%), Ustekinumab 1304 (5.70%); Tofacitinib, 308 (1.35%); Certolizumab, 248 (1.08%); and Golimumab, 121 (0.53%). There were 1780 (7.74%) patients with Nephrolithiasis: 438 (8.0%) patients were receiving biologic treatment. We found that the use of Vedolizumab (OR = 1.307, 95% CI 1.076–1.588, *p* = 0.0071) increased the odds of Nephrolithiasis by 31%. Conclusion: Vedolizumab use was associated with an increased risk of nephrolithiasis. The use of two or more biologic drugs also increased the risk compared to no biologic treatment.

## 1. Introduction

Nephrolithiasis is considered a health issue in several countries. In the Unites States, the incidence of nephrolithiasis is 5 to 15%, with 30 to 50% five-year recurrence [1]. Common risk factors for the development of kidney stones are gender and diet, with males having the highest incidence [1].

Chronic diseases with intermittent diarrhea and malabsorption, such as inflammatory bowel disease (IBD), which includes Crohn’s disease (CD) and ulcerative colitis (UC), are associated with the development of renal manifestations, specifically nephrolithiasis, tubulointerstitial nephritis, glomerulonephritis, and amyloidosis [2]. These complications that arise outside the intestinal inflammation of IBD are known as extraintestinal manifestations (EIMs) of IBD [3]. 

Biologic drugs are a class of medications that target specific molecules in the body’s immune system to control inflammation and reduce the symptoms of IBD [4]. Over the last two decades, and more recently, there has been an increase in the availability of biologics with different mechanisms of action for the treatment of CD and UC [4]. 

Recent studies have aimed to analyze the extent of renal manifestations in patients with IBD during the biologic era, suggesting a potential link between their use and an increased risk of nephrolithiasis [2]. We conducted the following study to further investigate the risk of nephrolithiasis in patients with IBD receiving biologic treatment. 

## 2. Materials and Methods

We performed a cross-sectional analysis of the Northwell Hospital Database between 1 January 2020 and 1 January 2022. Northwell hospitals include Glen Cove Hospital, Huntington Hospital, Lenox Health Greenwich village, Lenox Hill Hospital, LIJ Forest Hills, LIJ Valley Stream, Long Island Jewish, NSUH, Plainview Hospital, SIUH North and South, Southside Hospital and Syosset Hospital. Patients’ electronic charts were reviewed using the inclusion and exclusion criteria that are defined later. This study received institutional review board (IRB) approval (IRB #: 22-0794).

Inclusion criteria: Adult patients, 18 years and olderPatients diagnosed with IBD

Exclusion Criteria: 

Patients younger than 18 years 

Data collection was performed by searching inpatient and outpatient electronic databases using International Classification of Diseases 10 (ICD-10) codes. We used search parameters that included ulcerative colitis (UC), Crohn’s disease (CD), and inflammatory bowel disease (IBD). After identifying all IBD patients, we classified them based on whether they were receiving biologic therapy (Infliximab, Adalimumab, Certolizumab, Golimumab, Vedolizumab, Ustekinumab, Tofacitinib). Additionally, we used search parameters to detect the presence or absence of nephrolithiasis (ICD10: N20.0, N20.1, N20.2, N21.9, N21.0, N20.1, N20.8, N20.9, N22, N23) and gather patients’ baseline characteristics, alongside other known risk factors that might be potential confounders (i.e., gender, age, race, BMI, hyperparathyroidism, history of gastric bypass, diabetes mellitus, hypertension, gout, chronic kidney disease (CKD)) using International Classification of Diseases 10 (ICD-10) codes. The detection of the corresponding ICD 10 indicated the presence of the disease and the absence of the former indicated the absence of the latter. 

Continuous variables are reported as median (IQR), and categorical variables are reported as frequency (percent). The Mann–Whitney test was used to compare age between the groups of patients receiving any biologic treatment and those who were not receiving a biological treatment. Similarly, either the Chi-square test or the Fisher exact test was used for categorical variables. We used a multiple logistic regression model to assess whether an association existed between the use of biologic drugs and Nephrolithiasis, adjusting for several clinical confounders. We investigated the presence of a first-order interaction between the 7 biologic drugs, using a multiple logistic regression with a backward elimination procedure. We used an alpha of 0.01 for the interaction terms, due to the number of tests. As a sensitivity analysis, we fit a multiple logistic regression using the total number of biologic drugs (categories 0, 1, 2 and ≥3), rather than the number of individual drugs. The Hosmer–Lemeshow test was used to assess the goodness of fit for the logistic regression models. A two-sided *p* < 0.05 was considered statistically significant (except for the interaction terms, as described above). SAS 9.4 (SAS Institute Inc., Cary, NC, USA) was used for statistical analysis.

## 3. Results 

### Results and Outcomes

There were 22,895 patients aged ≥ 18. We excluded 11 patients who were receiving Natalizumab due to the small number. Additionally, there were 12 patients with two visits, who were also excluded. The final sample consisted of 22,860 patients with single visits. The median (IQR) age in the study was 55.0 (38.0–69.0) years old, and 2935 (52.38%) were females. (Table 1).

Among the 22,860 patients, 5603 (24.51%) were receiving at least one biologic drug. The biologic drugs received, from the most to the least commonly used, were as follows: Adalimumab, 2437 (10.66%); Infliximab, 1996 (8.73%); Vedolizumab 1397 (6.11%); Ustekinumab, 1304 (5.70%); Tofacitinib, 308 (1.35%); Certolizumab, 248 (1.08%); and Golimumab 121 (0.53%). 

Overall, 17,257 (75.49%) of the patients were not receiving biologic treatment, 3995 (17.48%) were receiving one biologic treatment, 1141 (4.99%) were receiving two biologic treatments, and 467 (2.04%) were receiving three or more biologic treatments. 

Our primary outcome was the presence of Nephrolithiasis (Y/N). Overall, there were 1780 (7.74%) patients with Nephrolithiasis: 448 (8.0%) patients were receiving biologic treatment, while there were 1332 (7.72%) patients not receiving biologic drugs. The percentage of stone formers in patients treated with each biologic is the following: Infliximab (7.62%), Adalimumab (8.33%), Certolizumab (10.08%), and Golimumab (9.09%).

We used a multiple logistic regression model to assess whether an association existed between the use of biologic drugs and Nephrolithiasis, adjusting for several clinical confounders (age, gender, race, ethnicity, body mass index, hyperparathyroidism, gastric bypass, hypertension, chronic kidney disease, and diabetes mellitus) (Table 2 and Table 3).

Adjusting for the confounders, we found that the use of Vedolizumab (OR = 1.307, 95% CI 1.076–1.588, *p* = 0.0071) increased the odds of Nephrolithiasis by 31% (95% CI 7.6–58.8%) (Table 3). 

Although none of the other biologic drugs (other than Vedolizumab) were statistically significant, the point estimates of the OR for all of them were close to or higher than 1. As a sensitivity analysis, we fit a multiple logistic regression using the total number of biologic drugs (categories 0, 1, 2 and ≥3), instead of the individual drugs (Table 4). The use of only one biologic drug (OR = 1.107, 95% CI 0.967–1.267, *p* = 0.1416) was not significantly associated with Nephrolithiasis, compared to no use of biologic drugs. The use of two drugs (OR = 1.254, 95% CI 1.008–1.559, *p* = 0.0424) and the use of three or more drugs (OR = 1.437, 95% CI 1.051–1.964, *p* = 0.0230) were associated with increasing the odds of Nephrolithiasis by 25.4% and 43.7%, respectively, relative to no use of biologic treatment. 

We investigated the presence of a first-order interaction between the seven biologic drugs using a multiple logistic regression with a backward elimination procedure. We used an alpha of 0.01 for the interaction terms due to the number of tests. None of the interactions had a *p*-value < 0.01, so we did not include the interaction terms. As a note, the *p*-value for the interaction of Adalimumab*Certolizumab was 0.02; this will need to be explored further in a larger study. In our sample, among the 94 patients using both drugs, 16 (17.02%) had Nephrolithiasis. 

## 4. Discussion

Urologic manifestations are one of the extraintestinal manifestations of IBD and they account for 22% of them [5]. They most commonly include enterovesical fistulas, ureteral obstruction, and nephrolithiasis [5], with the latter being the most common one [6]. Previous studies have established that the prevalence of kidney stones is higher in IBD patients than in the general population, especially in CD patients [5]. It is present in around 7–15% of patients and more pronounced, with small bowel resection or continuous inflammation [1]. 

The medical therapies for Crohn’s disease (CD) and ulcerative colitis (UC) have expanded rapidly over the last two decades, and more recently, there has been an increase in the availability of biologics with different mechanisms of action [4]. No studies exist to exclusively analyze the effect of biologic therapy on nephrolithiasis. For Cury et al., although any medication use was associated with nephrolithiasis, it did not reach statistical significance when controlled for the activity of the disease [1]. Abdulrahman et al. conducted a systematic review and meta-analysis that included 13,339,065 patients in order to assess the risk factors of nephrolithiasis in IBD patients. TNF-α (Tumor Necrosis Factor) inhibitors were not associated with nephrolithiasis in IBD patients [7].

Our study aims to investigate the impact of biologics on nephrolithiasis. The prevalence of nephrolithiasis overall in our study was 7.71%, which is similar to the literature [1]. 

There are several risk factors for nephrolithiasis in IBD. Diarrheal illness is one of the factors shared by both CD and UC, increasing the risk of nephrolithiasis [3]. Other risk factors include younger age, male sex, and gastrointestinal surgery [8,9]. These risk factors were also found to be associated with renal stones in our study. Secondary hyperparathyroidism is frequently found in IBD, unlike its primary form [10]. In our study, hyperparathyroidism was also associated with nephrolithiasis. Other risk factors for urolithiasis include citrate deficiency, hyperuricosuria, and hypercalciuria [11]. Mihai et al. conducted a randomized control trial to understand the effects of citrate therapy on the stone-free rate after 90 days in patients who underwent flexible digital ureteroscopy with laser lithotripsy. The study concluded that the addition of citrate therapy improved the stone-free and stone-expulsion rate [11].

In mild to moderate forms of IBD, Aminosalicylates, or drugs that contain 5ASA (Aminosalicylic acid), are considered one of the traditional treatments implemented in such forms. Sulfasalazine and mesalazine belong to this group [12]. Although sulfasalazine has a less favorable side effect profile than its counterpart, mesalazine is associated with several renal manifestations, including nephrolithiasis [13]. 

Additionally, immunomodulators are another class of medications that can be used in IBD. These include 6 MP (mercaptopurine) and AZT (Azathioprine) [12]. In Abdulrhman et al.’s systematic review and meta-analysis, there was not an association between nephrolithiasis and both 6MP and AZT [7].

Biologics are a class of drugs that target pro-inflammatory cytokines, such as TNF-α, IL-12, IL-23, and integrin [12]. In patients with moderate to severe IBD, more specifically UC, Vedolizumab is considered one of the first-line therapies [14]. Vedolizumab is a humanized anti-alpha-4-beta-7 integrin monoclonal antibody that has been found to be safe and effective in IBD patients, and minimize the risk of renal side effects [15]. In a retrospective study conducted by Dubinsky et al., patients on vedolizumab were at a higher risk of developing EIM than patients on anti-TNF [16]. Although case reports have been found linking vedolizumab to acute interstitial nephritis [15], no association has been found between the former and nephrolithiasis. However, in our study, we found a 31% increase in the odds of developing nephrolithiasis while on vedolizumab. 

Interestingly, our study found that the use of two drugs and three or more drugs was associated with increases of 25.4% and 43.7% in the odds of experiencing nephrolithiasis, respectively. This, however, could be attributed to the severity of IBD that necessitated that patients be on multiple lines of biologic therapies in order to control its activity. In fact, Dincer et al. established an association between patients receiving anti-TNF therapy for IBD and a higher risk of developing renal manifestations [2]. This was attributed to the fact that nephrolithiasis, the predominant renal manifestation, was associated with a more severe IBD activity, consequently needing biologic treatment to achieve better control [2].

The early detection of kidney stones in IBD patients is essential, as recurrence is associated with CKD and ESRD [6,17]. Thus, it could be of benefit to establish guidelines or expert recommendations that are able to better monitor the development of nephrolithiasis in at-risk IBD patients who are starting to receive biologic therapy. 

Our study is not without limitations. First, the reliance on ICD-10 codes for identifying patient diagnoses may have led to coding errors. Second, we were unable to associate the presence of nephrolithiasis with disease activity. Third, we were unable to account for a prior history of nephrolithiasis. Additionally, we did not differentiate primary forms of hyperparathyroidism from secondary forms. Finally, we do not know whether the detection of nephrolithiasis was achieved incidentally or due to a symptomatic presentation of renal colic. 

However, our research is strengthened by taking into consideration the impact of biologic therapy on nephrolithiasis in patients with inflammatory bowel disease, despite the limited guidance available in the existing literature. 

## 5. Conclusions

The study found that vedolizumab use was associated with an increased risk of nephrolithiasis, and that the use of two or more biologic drugs also increased the risk compared to no biologic treatment. However, the use of only one biologic drug was not significantly associated with nephrolithiasis. Patients with IBD considering biologic therapy should discuss its potential risks and benefits with their healthcare provider and undergo regular monitoring for kidney stone formation. Further research is needed to understand the specific mechanism behind these findings.

## Figures and Tables

**Table 1 jcm-12-06114-t001:** Baseline characteristics of the study population.

Variable		Biologic Use		*p*-Value
		No = 17,257	Yes = 5603	
Age	25th Pctl	41.0	31.0	<0.0001
	50th Pctl	58.0	46.0	
	75th Pctl	71.0	61.0	
Gender	Female	9964 57.74%	2935 52.38%	<0.0001
	Male	7293 42.26%	2668 47.62%	
Race	Asian	592 3.43%	212 3.78%	0.0001
	Black	1360 7.88%	395 7.05%	
	Other	1875 10.87%	568 10.14%	
	Unknown	1206 6.99%	316 5.64%	
	White	12,224 70.84%	4112 73.39%	
Ethnicity	Hispanic or Latino	1306 7.57%	332 5.93%	<0.0001
	Non-Hispanic or Latino	13,988 81.06%	4709 84.04%	
	Unknown	1963 11.38%	562 10.03%	
BMI	Underweight	652 3.78%	250 4.46%	<0.0001
	Normal	5652 32.75%	1989 35.50%	
	Overweight	5005 29.00%	1642 29.31%	
	Obese	4120 23.87%	1357 24.22%	
	Unknown	1828 10.59%	365 6.51%	
Hyperparathyroidism	No	17,022 98.64%	5544 98.95%	0.0021
	Yes	235 1.36%	59 1.05%	
Gastric Bypass	No	14,548 84.30%	5050 90.13%	0.0747
	Yes	2709 15.70%	553 9.87%	
DM	No	10,419 60.38%	4151 74.09%	<0.0001
	Yes	6838 39.62%	1452 25.91%	
HTN	No	17,022 98.64%	5544 98.95%	<0.0001
	Yes	235 1.36%	59 1.05%	
CKD	No CKD	16,229 94.04%	5431 96.93%	<0.0001
	Stage 5	75 0.43%	11 0.20%	
	Stage < 5	953 5.52%	161 2.87%	

CKD: Chronic kidney Disease, DM: Diabetes Mellitus, Hyperpara: Hyperparathyroidism, HTN: Hypertension, Pctl: Percentile. ‘CKD5′ is selected if the corresponding variable is selected, regardless of any other variables. ‘CKD < 5′ is the category if any of the other CKD variables [CKD2–4] or the general CKD variable are selected, regardless of any conflicts. ‘No CKD’ is the category if none of the CKD variables are selected.

**Table 2 jcm-12-06114-t002:** Odds Ratios (OR) from the multiple logistic regression model, using the individual biologic drugs.

Effect		Estimate	95% Confidence Limits		*p*-Value
Age		1.006	1.003	1.009	0.0003
Gender	Male vs. Female	1.628	1.472	1.801	<0.0001
Race	Asian vs. White	0.533	0.383	0.742	0.0002
	Black vs. White	0.465	0.367	0.589	<0.0001
	Other vs. White	0.685	0.559	0.840	0.0003
	Unknown vs. White	0.787	0.569	1.090	0.1490
Ethnicity	Hispanic or Latino vs. Non-Hispanic or Latino	1.378	1.108	1.713	0.0039
	Unknown vs. Non-Hispanic or Latino	0.579	0.447	0.748	<0.0001
Hyperpara.		2.795	2.023	3.862	<0.0001
G-Bypass		1.425	0.987	2.059	0.0589
DM		1.305	1.144	1.488	<0.0001
HTN		1.415	1.251	1.601	<0.0001
ESRD		1.862	1.069	3.244	0.0282
CKD		1.682	1.410	2.007	<0.0001
BMI	Underweight	1.061	0.814	1.382	0.6614
	Overweight	1.099	0.970	1.246	0.1382
	Obese	1.241	1.089	1.413	0.0012
	Unknown	0.460	0.349	0.606	<0.0001

BMI: Body Mass Index, CKD: Chronic Kidney Disease, DM: Diabetes Mellitus, ESRD: End-Stage Renal Disease, HTN: Hypertension.

**Table 3 jcm-12-06114-t003:** Odds Ratios (OR) from the multiple logistic regression model, using the individual biologic drugs.

Effect	Estimate	95% Confidence Limits		*p*-Value
Adalimumab	1.138	0.967	1.338	0.1197
Infliximab	1.022	0.852	1.227	0.8112
Vedolizumab	1.307	1.076	1.588	0.0071
Ustekinumab	1.011	0.811	1.259	0.9246
Tofacitinib	1.085	0.723	1.629	0.6946
Certolizumab	1.336	0.869	2.065	0.1872
Golimumab	0.951	0.501	1.803	0.8766

**Table 4 jcm-12-06114-t004:** Odds Ratios (OR) from the multiple logistic regression model, using the total number of biologic drugs.

Effect		Estimate	95% Confidence Limits		*p*-Value
Age		1.006	1.003	1.010	0.0003
Gender	Male vs. Female	1.627	1.471	1.799	<0.0001
Race	Asian vs. White	0.530	0.381	0.738	0.0002
	Black vs. White	0.462	0.365	0.585	<0.0001
	Other vs. White	0.685	0.559	0.840	0.0003
	Unknown vs. White	0.786	0.568	1.088	0.1466
Ethnicity	Hispanicor Latino vs. Non-Hispanicor Latino	1.374	1.105	1.709	0.0042
	Unknown vs. Non-Hispanicor Latino	0.580	0.448	0.750	<0.0001
BMI	Underweight	1.063	0.816	1.384	0.6532
	Overweight	1.100	0.970	1.247	0.1366
	Obese	1.241	1.089	1.413	0.0012
	Unknown	0.461	0.350	0.608	<0.0001
Hyperparathyroidism		2.785	2.016	3.848	<0.0001
Gastric Bypass		1.427	0.988	2.061	0.0580
DM		1.307	1.146	1.490	<0.0001
HTN		1.414	1.251	1.600	<0.0001
ESRD		1.871	1.074	3.259	0.0269
CKD		1.681	1.410	2.005	<0.0001
Number of biologics	1	1.107	0.967	1.267	0.1416
	2	1.254	1.008	1.559	0.0424
	3 or more	1.437	1.051	1.964	0.0230

BMI: Body Mass Index, CKD: Chronic Kidney Disease, DM: Diabetes Mellitus, HTN: Hypertension.

## Data Availability

The data that support the findings of this study are available from Northwell Health Database.

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
