# Peer review of "Risk of Nephrolithiasis in Patients with Inflammatory Bowel Disease Receiving Biologic Treatment"

_jcm, 2023, doi:10.3390/jcm12196114_

Round 1

Reviewer 1 Report

Expand the abbreviations on first mention in the text (UTI, CKD) and table 1 DM HTN, table 2 ESRD

Lines 92-96 seem to be repeated in 131-135
delete 89-91, subsection 3.1. is repeated.

Chceck guidelines for authors in order to learn how to introduce tables in the article

Line 242 seems to be included by mistake

Not many references are included in this paper. Consider citing the latest study Abdulrhman A, Alsweed A, Alotaibi MR, Aldakhil AY, Alahmadi SF, Albishri SM, Alhmed NI. Urolithiasis in patients with inflammatory bowel disease: A systematic review and meta-analysis of 13,339,065 individuals. Medicine (Baltimore). 2023 Jun 16;102(24):e33938.

Author Response

We thank the reviewer for his comments

  • Expansion of the abbreviations completed
  • To note that the history of recurrent UTI was removed due to the unavailability of data. we apologize for the confusion
  • Repetition was removed.
  • The tables' introduction was altered to conform with the author's guidelines. 
  • Line 242 was removed.
  • The citation was added for Abdulrahman et al.'s systemic review:  Abdulrahman et al. conducted a systematic review and meta-analysis that included 13,339,065 patients in order to assess the risk factors of nephrolithiasis in IBD patients. TNF-α (Tumor Necrosis Factor) inhibitors were not associated with nephrolithiasis in IBD patients. 
  • Other references were added regarding the treatment of IBD and its association with nephrolithiasis: In mild to moderate forms of IBD, Aminosalicylates or drugs that contain 5ASA (Aminosalicylic acid) are considered one of the traditional treatments implemented in such forms. Sulfasalazine and mesalazine belong to this group.12 Although sulfasalazine has a less favorable side effect profile than its counterpart, mesalazine is associated with renal manifestations including nephrolithiasis.13 Additionally, immunomodulators are another class of medications that can be used in IBD. These include 6 MP (mercaptopurine) and AZT (Azathioprine). In Abdulrhman et al.’s systematic review and meta-analysis, there wasn’t an association between nephrolithiasis and both 6MP and AZT.7 Biologics are a class of drugs that target pro-inflammatory cytokines, such as TNF-α, IL-12, IL-23, and integrin.12

Reviewer 2 Report

Dear Authors,

I read with great interest your article about using biologics and the prevention of nephrolithiasis.

However, there are some aspects that need improvement.

Although, you are natural English language speakers please ask a colleague to reread the manuscript for some small typing errors.

At the end of the manuscript you mention supplementary materials which are not available. Either remove that section or make the supplementary materials available.

Moreover,  remove the section with the patents.

Furthermore, in the discussion section you should speak about other causes of nephrolitiasis and types of preventing this associated pathology. Please reference this to the work: STONE FREE-RATE EXPERIENCE IN POST-INTERVENTIONAL ... https://farmaciajournal.com/wp-content/uploads/art-16-Ene_Geavlete_573-580.pdf

You need to improve the number of references in order to gain attention to the article.

In the end try to expand on the limitation of your study.

Looking forward to receiving the improved manuscript.

Author Response

We thank the reviewer for his comments

  • Typing errors were corrected.
  • The supplementary materials section was removed, we apologize for the confusion 
  • We added a section regarding other causes of nephrolithiasis and types of prevention: Other risk factors for urolithiasis include citrate deficiency, hyperuricosuria, and hypercalciuria.11 Mihai et al. conducted a randomized control trial to understand the effects of citrate therapy on the stone-free rate after 90 days in patients who underwent flexible digital ureteroscopy with laser lithotripsy. The study concluded that the addition of citrate therapy improved the stone-free and stone-expulsion rate.11
  • Other references were added regarding the treatment of IBD and its association with nephrolithiasis in order to increase the attention as suggested: Abdulrahman et al. conducted a systematic review and meta-analysis that included 13,339,065 patients in order to assess the risk factors of nephrolithiasis in IBD patients. TNF-α (Tumor Necrosis Factor) inhibitors were not associated with nephrolithiasis in IBD patients.7 In mild to moderate forms of IBD, Aminosalicylates or drugs that contain 5ASA (Aminosalicylic acid) are considered one of the traditional treatments implemented in such forms. Sulfasalazine and mesalazine belong to this group.12 Although sulfasalazine has a less favorable side effect profile than its counterpart, mesalazine is associated with renal manifestations including nephrolithiasis.13 Additionally, immunomodulators are another class of medications that can be used in IBD. These include 6 MP (mercaptopurine) and AZT (Azathioprine). In Abdulrhman et al.’s systematic review and meta-analysis, there wasn’t an association between nephrolithiasis and both 6MP and AZT.7 Biologics are a class of drugs that target pro-inflammatory cytokines, such as TNF-α, IL-12, IL-23, and integrin.12
  • We added to the limitation: Additionally, we did not differentiate primary from secondary forms of hyperparathyroidism.

Round 2

Reviewer 1 Report

I do not think the Authors applied all the remarks in the final version of the manuscript. For example lines 90-91 should be deleted. Section results still have repetitions in 3.1.

Author Response

we thank the reviewer for his comments

Upon review of the newly submitted revised version of the manuscript, I could not find the repetitions but we are happy to fix them if you can kindly point them out to us. apologies for any inconvenience.